# Investigating Differential Expressed Genes of *Limosilactobacillus reuteri* LR08 Regulated by Soybean Protein and Peptides

**DOI:** 10.3390/foods11091251

**Published:** 2022-04-26

**Authors:** Shuya Zhu, Yinxiao Zhang, Jingyi Wang, Chi Zhang, Xinqi Liu

**Affiliations:** Beijing Advanced Innovation Center for Food Nutrition and Human Health, Beijing Engineering and Technology Research Center of Food Additives, National Soybean Processing Industry Technology Innovation Center, School of Food and Health, Beijing Technology and Business University, Beijing 100048, China; btbuzhushuya@163.com (S.Z.); zhangyx268825@126.com (Y.Z.); wjy18920879790@163.com (J.W.)

**Keywords:** soybean protein, soybean peptides, *L. reuteri*, transcriptomics, purine metabolism, ABC transporters, fatty acid biosynthesis

## Abstract

Soybean protein and peptides have the potential to promote the growth of *Lactobacillus*, but the mechanisms involved are not well understood. The purpose of this study is to investigate differentially expressed genes (DEGs) of *Limosilactobacillus reuteri* (*L. reuteri*) LR08 responding to soybean protein and peptides using transcriptome. The results showed that both digested protein (dpro) and digested peptides (dpep) could enhance a purine biosynthesis pathway which could provide more nucleic acid and ATP for bacteria growth. Moreover, dpep could be used instead of dpro to promote the ABC transporters, especially the genes involved in the transportation of various amino acids. Interestingly, dpro and dpep played opposite roles in modulating DEGs from the *acc* and *fab* gene families which participate in fatty acid biosynthesis. These not only provide a new direction for developing nitrogen-sourced prebiotics in the food industry but could also help us to understand the fundamental mechanism of the effects of dpro and dpep on their growth and metabolisms and provides relevant evidence for further investigation.

## 1. Introduction

*Limosilactobacillus reuteri* (*L. reuteri*), as one of the most important probiotics, has been gradually used in the functional food and dairy industry [1]; this is because it colonizes the gastrointestinal tract and is resistant to gastric acid and bile salts, thereby benefitting human health by manipulating the gut microflora [2]. It has been reported that *Lactobacillus* have various beneficial effects including amelioration of symptoms of lactose intolerance [3] and the reduction of the risk of necrotizing enterocolitis [4], and that some strains can also secrete bacteriocin to inhibit the growth of pathogens in the intestinal tract [5] or improve the immunity of the human body to inhibit the development of pathogens; therefore, they can positively affect human health [6,7].

Food proteins and their hydrolysates also have many bioactivities, such as antihypertension, reducing cholesterol, antioxidant, etc. [8,9,10]. Recent studies have reported that protein hydrolysates and peptides can specifically promote the growth and metabolism of *Lactobacillus*. Researchers found that whey protein hydrolysate can promote the proliferation of *Lactobacillus acidophilus* [11]. Others also found that the pea protein hydrolysate significantly promotes the growth of probiotics and can improve the survival rate of probiotics [12]. These characteristics could improve their ability to inhibit pathogens. Additionally, some antimicrobial peptides could inhibit pathogens directly [13,14]. However, these studies still focus on exploring the relationship between proteins from different sources and different kinds of probiotics. In addition, our previous study has shown that both soybean protein and soybean peptides, as excellent protein resources, could also play a crucial role in *L. reuteri* growth and metabolism and were highlighted as potential prebiotic candidate [15,16]. Whilst the mechanism of their effects was still inadequate, they nevertheless warrant further investigation. Transcriptome was chosen as the first step since it plays a critical role in exploring the overall gene expression in *Lactobacillus* in order to understand the preliminary mechanisms at RNA level [17].

The purpose of this article is to investigate the preliminary mechanism of soybean protein and soybean peptides affecting the growth and metabolism of *L. reuteri* LR08. The results can provide a theoretical basis for discovering a nitrogen source with potential probiotic function for prebiotic families as well as exploring the mechanism of protein and peptides on probiotics.

## 2. Materials and Methods

### 2.1. The Preparation and Properties of Soybean Protein and Peptide Hydrolysates

The sources of original soybean protein and peptides were consistent with those described by Zhao [18]. Soybean protein and peptides hydrolysates were prepared according to the previous method of simulating gastrointestinal digestion in vitro [19]. Each sample was dissolved in distilled water (10 g/L) and pepsin was added after the pH was adjusted to 2.0 with 6 N HCl (enzyme to substrate ratio of 1:35 *w*/*w*). The mixture was stirred at 37 °C for 1 h. The pH was then adjusted to 5.3 with a saturated NaHCO_3_ solution and further to pH 7.5 with 5 N NaOH. Pancreatin was added to the mixture (enzyme to substrate ratio of 1:25 *w*/*w*), followed by incubation at 37 °C for 2 h. After proteolysis, the reaction was terminated by heating to 95 °C for 10 min, and the supernatant was lyophilized for 36 h and stored at −20 °C for future analysis. The nitrogen content of digested soybean protein (dpro) and digested soybean peptide (dpep) was 10.93 g/100 g and 10.14 g/100 g, respectively. The molecular weight of dpro was lower than 35 kDa, and dpep was mostly lower than 500 Da (71.65%) [15].

### 2.2. Activation and Growth Condition of Strains

*L. reuteri* was obtained from China General Microbiological Culture Collection Center (CGMCC). The MRS, dpro, and dpep treatments were prepared according to Zhang’s method [15]. The *L. reuteri* bacteria stored in the refrigerator at −80 °C were activated in 3 mL MRS medium, and then cultured at 37 °C for 12 h. Next, 200 microliters of culture solution were inoculated in 50 mL MRS medium for 12 h at 37 °C for follow-up experiments under different treatments. After being incubated for 10 h with different treatments, samples of *L. reuteri* (the logarithmic growth phase of *L. reuteri* LR08) were selected as samples for transcriptome analysis. MRS was used as the positive control group, and digested soybean protein and digested soybean peptides were added into the culture medium, replacing half of the nitrogen content (i.e., dpro group and dpep group). The compositions of related culture medium are shown in Table 1.

### 2.3. Library Construction for Transcriptome Sequencing

Total RNA of *L. reuteri* LR08 was extracted using a Bacterial Total RNA Extraction Kit (Tiangen Biotech Co., Ltd., Beijing, China). The RNA-seq transcriptome library was constructed using Illumina TruSeqTM RNA Sample Prep Kit. The mRNA was separated by oligo-dT magnetic beads. First, fragmentation was performed with fragment buffer, and then double-stranded cDNA was synthesized using SuperScript double-stranded cDNA synthesis kit (Invitrogen, Carlsbad, CA, USA). The target cDNA fragment of 200 bp–300 bp was selected after completing the repaired end of the synthesized cDNA. A total of 15 cycles of PCR amplification were conducted using Phusion DNA polymerase (NEB). The two-terminal RNA-seq sequencing library for computer sequencing was performed using Illumina Novaseq 6000 (2 × 150 bp) after TBS380 quantification.

### 2.4. Reverse Transcription of RNA

After total RNA of *L. reuteri* LR08 extraction, the reverse transcription was conducted for preparing cDNA with the help of a Reverse Transcription Kit (Tiangen, Beijing, China). The specific method was as follows: the 5×FastKing-RT SuperMix and RNase-Free ddH_2_O was thawed at room temperature (15–25 °C) and was then quickly put on ice. Each solution was blended evenly using vortex oscillation before being prepared for use and centrifuged briefly to collect the liquid remaining on the pipe wall. The reverse transcription reaction system that consisted of 4 μL 5× FastKing-RT SuperMix, 1 μg total RNA, and 15 μL RNase-Free ddH_2_O was configured on ice, and then the reaction was carried out. The genome and reverse transcription reaction were removed by reacting for 15 min at 42 °C, and then the enzyme activity was inactivated for 3 min at 95 °C. The resulting cDNA solution was stored at −80 °C for subsequent use.

### 2.5. Fluorescence Quantitative PCR Reaction

The reaction of qPCR was performed following the program of “95 °C for 10 s for denaturation, 63 °C for 20 s for annealing, and 72 °C for 32 s for extension” using a SuperReal PreMix Plus Kit (TransGen, Beijing, China). Primers sequences were designed using Beacon Designer 8.0 software (Bomeaide Gene Technology, Beijing, China) for 16 crucial differentially expressed genes, and the sequence of the internal reference gene *Lpla16rsRNA* was referred to [20] (Appendix A). The 2^−ΔΔt^ method was used to evaluate the fold change with relation to the control group of selected genes.

### 2.6. Statistical Analysis

SeqPrep and Sickle software were used to perform the clipping and quality control of primary end-numbers with default parameters. TopHat software was used to compare with the reference genome in the orientation mode Clean reads. The Mapped Reads were spliced using StringTie software, based on the selected reference genome sequence, and then compared with the original genome annotation information. Statistical analysis was carried out for Raw counts using DESeq2 software, based on *p* < 0.05 and |log2FC| ≥ 2 as the screening conditions, and genes with different expressions between the experimental group and the blank group were screened. Excel 2016 was used to analyze the variance of experimental data which were expressed as mean ± standard deviation and to draw a graph. * *p* < 0.05, ** *p* < 0.01 or *** *p* < 0.001 was statistically significant. The analysis of variance (ANOVA) was performed a as one-way method for the analysing the data. Tukey–Kramer and Games–Howell tests were performed to compare the groups’ mean values.

## 3. Results and Discussion

### 3.1. Overview of the L. reuteri LR08 Transcriptomic Changes Response to dpep and dpro

The transcriptomic analysis was performed to deepen the understanding of overall gene changes of *L. reuteri* LR08 when reacted with dpep and dpro. The three groups of sample clusters were analyzed by principal component analysis (PCA) and correlation analysis methods [21]. In Figure 1, samples of MRS, dpro, and dpep treatment are concentrated in each group, with obvious separation between groups. As can be seen from Figure 2, the correlation coefficient between any two samples within the group is above 0.9, which indicates that the three samples have high biological repeatability and the experimental design is reasonable. This indicates reliable research results which can be used for subsequent analysis. A total of 380 DEGs are screened in the dpep/MRS group, among which 206 genes are up-regulated and 174 genes are down-regulated. A total of 401 DEGs are screened in the dpro/MRS group, among which 276 genes are up-regulated and 125 genes are down-regulated. There are 276 DEGs found in the dpep/dpro group, with 103 up-regulated genes and down-regulated 173 genes (Appendix A as Appendix A). While comparing up-regulated genes in three comparisons, dpep/MRS and dpro/MRS dpep/MRS group have 77 identical ones, which mainly focus on genes related to purine metabolism, and most of these genes have higher up-regulation folds in the dpro/MRS group. The dpep/MRS group and the dpep/dpro group have 58 identical upregulated genes, which mainly focus on fatty acid biosynthesis and pyruvate metabolism. Among down-regulated DEGs of three comparisons, dpep/MRS and dpro/MRS have 68 identical ones, which are mainly related to ribosomal proteins, pentulose and hexulose kinase, lysine, isoleucine, methionine and threonine biosynthesis process, and TCA cycle, etc. There are 33 down-regulated genes in the dpep/MRS group and dpep/dpro group, most of which are related to glutamyl-tRNA biosynthesis, ABC transporters, and ammonium transporter.

### 3.2. Gene Ontology (GO) Enrichment Analysis

Compared with MRS, dpep elevated transcription of genes mainly enriched in. the biological process (BP) of IMP biosynthesis and purine biosynthesis, which is critical for cell division during bacteria growth (Figure 3A). Organonitrogen compound biosynthetic processes, including various amino acids and peptide biosynthetic process, were expectedly downregulated. Additionally, within the molecular function (MF) category, amino acid-transporting ATPase activity was also inhibited, given that dpep contains less free amino acids for bacteria to take advantage. The distribution of up-regulated DEGs in the BP category of the dpro/MRS comparison group was similar to the dpep/MRS group (Figure 3B), while those in the MF classification—which were principally involved in regulating the activities of phosphoribosylformylglycinamidine synthase, acetyl-CoA carboxylase, ligase forming carbon-carbon bonds, and CoA carboxylase—were different. When the dpep group was compared with the dpro group (Figure 3C), DEGs were especially induced and enriched in the BP of nicotinamide adenine dinucleotide (NAD) and proton-transporting ATP synthase in catalytic core F(1), proton-transporting two-sector ATPase complex in catalytic domain, and acetyl-CoA carboxylase within cellular component (CC) category, which could promote the increase of intracellular ATP level for cell division and growth. Furthermore, it was found that soybean peptides were more effective in contributing *L. reuteri* growth than both the MRS and dpro groups.

### 3.3. Kyoto Encyclopedia of Genes and Genomes (KEGG) Enrichment Analysis

As can be seen in Figure 4, there were 11 identical gene items in the dpro/MRS group and dpep/MRS group. In the dpro/MRS comparison, purine metabolism, and fatty acid biosynthesis were significantly enriched, while in the dpep/MRS group, ABC transport, fatty acid biosynthesis, purine metabolism, and ribosome were significantly enriched. Although these two groups were similar in purine metabolism items, they were different in terms of the degree of enrichment. From the perspective of dpro/MRS comparison, there were 20 genes enriched in purine metabolism, of which 15 were up-regulated and 5 were down-regulated. However, there were 17 DEGs, of which 13 were up-regulated and 4 were down-regulated in the dpep/MRS group. DEGs in the dpep/dpro group were mainly enriched in oxidative phosphorylation, fatty acid biosynthesis, photosynthesis, ABC transporters, pyruvate metabolism, and citrate cycle (TCA cycle). Although ABC transporters were enriched in both the dpep/MRS group and dpep/dpro group, the number of genes was also different. The dpep/MRS group contained 20 genes (15 up-regulated and 5 down-regulated). In the dpep/dpro group there were only 15 genes (11 up-regulated and 4 down-regulated). The fatty acid biosynthesis pathway containing five identical genes was significantly enriched among the three comparison groups. It can be implied that, compared with MRS, dpro and dpep can regulate purine metabolism and fatty acid biosynthesis to regulate energy metabolism of *L. reuteri*, and that dpep can also regulate nutrients absorption and protein production of bacteria through ABC transporters and ribosome. Compared with dpro, dpep can also regulate the energy metabolism of *L. reuteri* through genes’ enrichment in oxidative phosphorylation, pyruvate metabolism, and citrate cycle.

### 3.4. Analysis of Critical DEGs from GO and KEGG under dpep and dpro Treatment

#### 3.4.1. DEGs Enriched in Fatty Acid Biosynthesis Pathway Were Oppositely Modulated by dpep and dpro

DEGs were significantly enriched in a fatty acid biosynthesis pathway both in the dpro/MRS group and dpep/MRS group, and there were five enriched genes, namely *accA* (*N134_RS05255*), *accC* (*N134_RS05265*), *accD* (*N134_RS05260*), *fabI* (*N134_RS05250*), and *fabZ* (*N134_RS05270*). They were significantly upregulated in the dpep/MRS group and dpep/dpro group (Figure 5), but downregulated in the dpro/MRS group. Acetyl-CoA carboxylase carboxyltransferase subunit β (*accD*) [22], ATP-grasp domain-containing protein (*accC*), and acetyl-CoA carboxylase carboxyl transferase subunit α (*accA*) [23] belong to the group of carboxylases which are essential for the first step of fatty acid synthesis reaction using biotin as a cofactor [24]. The upregulated *acc* gene family could enhance the carboxylation of acetyl-CoA, providing enough malonyl-CoA serves as a two-carbon extender unit for the synthesis of fatty acids. β-hydroxacyl-ACP-dehydratase (*fabZ*, *N134_RS05270*) and enoyl-ACP-reductase (*fabI*, *N134_RS05250*) are closely related in the subsequent fatty acid prolongation reaction [25]. *FabZ* regulates the activity of 3-hydroxyacyl-ACP dehydratase, which is the dehydration step in the elongation of fatty acids promoting the conversion of 3-hydroxyacyl-ACP to enoyl-ACP [26]. Therefore, the increase expression of *fabI* and *fabZ* would enhance unsaturated fatty acid synthesis, which might alter the composition of fatty acids in plasma membrane. It has been observed that the increased level of unsaturated fatty acid synthesis in *Lactobacillus helveticus* prevented major damage to viability in cells exposed to the various stress combinations, thereby providing survival advantages [27]. This may explain the phenomenon of soybean peptides promoting the viable cell count of *L. reuteri* in our previous investigation [28]. In addition, the up-regulation of DEGs in fatty acid biosynthesis may promote the secretion of short-chain fatty acids (SCFA), which is consistent with our previous findings that dpep, compared with dpro, can stimulate *L. reuteri* to produce more organic acids in metabolism. Conversely, the downregulated expression of *fabZ* and *fabI* affected by soybean protein might not be conducive to the growth of *L. reuteri*, relatively.

#### 3.4.2. Both dpep and dpro Improve Gene Expression Enriched in Purine Biosynthesis

DEGs in both dpep and dpro treatment were, compared with MRS group, significantly enriched in purine metabolism pathway, within which a series of DEGs involved in purine biosynthesis were upregulated (Figure 5). The expression of *purQ* which encodes the glutaminase domain increased most, with 47.09 folds, in the dpep group. *PurQ* form complex with *purS* and *purL* [29], which were also highly induced, catalyzes the fourth step of the IMP biosynthesis responsible for the catalysis of ammonia formation using glutamine. *PurN* encoded phosphoribosyl glycinamide formyltransferase was also induced relatively high expression with 36.83 folds, which catalyzes the third step of IMP synthesis [30]. Additionally, *purF*, *purD*, *purM*, *purC*, *purH*, etc., were upregulated significantly, which almost covers all critical steps of ten procedures of synthesis of IMP. IMP could be subsequently converted to GMP and AMP, which not only play a critical role in the synthesis of RNA and DNA [31], but also are pivotal ingredients of ATP and GTP synthesis. These results suggested soy peptides and soy protein may promote nucleic acid and energy-rich phosphate compounds synthesis, so that it can provide *L. reuteri* LR08 with enough available energy and growth materials for their development. Other sourced protein such as bovine lactoferrin [32] and whey [33] were also explored in terms of their influence on the expression changes of purine biosynthesis by transcriptome analysis. Whey barely affected the relative genes of purine synthesis in *Lactobacillus rhamnosus* GG (LGG). Interestingly, bovine lactoferrin even inhibited purine synthesis of LGG, which may be related to the diversity of composition of various protein and strain of *Lactobacillus*.

#### 3.4.3. Dpep Induced Genes Enriched in ABC Transporters Pathway

ABC transporter protein is one of the largest superfamilies found in bacteria, which is responsible for cells to absorb essential nutrients and transport molecules from outside cells to organelles [34]. Compared with soybean protein, DEGs coding for some ABC-type transporters mainly associated with various amino acids transportation showed marked expression changes in response to dpep treatment. *TcyA* (*N134_RS01535*), *tcyB* (*N134_RS01525*), and *tcyC* (*N134_RS01530*) encoding transporter substrate-binding domain-containing protein [35], amino acid ABC transporter permease, and amino acid ABC transporter ATP-binding protein, which jointly participated in cysteine transportation, exhibited the relatively high up-regulation folds among all the upregulated genes in ABC transporters pathway. Additionally, genes for D-methionine (*N134_RS08600*) [36] and glycine (*N134_RS00920*) also increased by relatively high folds, while the *glnH* (*N134_RS00495*) and *glnQ* (*N134_RS00500*) responsible for glutamine transport [37] and the *artQ* (*N134_RS04200*) and *artR* (*N134_RS04205*) responsible for arginine lysine or histidine transport [38] were significantly inhibited (Figure 5). Up-regulation and down-regulation of these transporters are probably to promote and prevent accumulation of these amino acids inside the cell, suggesting that digested soybean peptides may contain less cysteine, D-methionine, and glycine, and more glutamine, arginine, or histidine. From another point of view, soybean peptides may improve the nutrient absorption by increasing the expression of the genes related to some kinds of amino acids ABC transporters, thus promoting the growth of *L. reuteri* LR08. However, the expression of these genes has no significant difference in dpro/MRS comparison, probably since that it is easier and more direct for *L. reuteri* to use smaller molecule weights of nitrogen sources. The expression of *opuC*, *opuBD*, and *opuA* as transport system proteins of osmoprotectants also increased [39]. Glycine betaine is the main osmoprotectant in bacteria [40] and can be produced by the gradual methylation of glycine. Its input and high-level accumulation helped bacterial cells to cope with the increase of external osmotic pressure by balancing the osmotic gradient across the cytomembrane [41]. This may also explain why *L. reuteri* still showed strong growth under the increased osmotic pressure, despite higher accumulation of metabolites in dpep. A similar result was also elucidated in LGG under bovine lactoferrin treatment [32]. Moreover, Li et al. [42] found mulberry galacto-oligosaccharide could induce ABC transporters responsible for both carbohydrates and nitrogens.

### 3.5. Real-Time Fluorescence Quantitative PCR (RT-qPCR) Verification

16 genes in Appendix A were verified by qPCR in order to verify the accuracy of RNA-seq with Lpla16rsRNA as the internal reference gene. Of the 48 samples in the 3 groups, 29 genes had the same expression pattern as RNA-seq results, accounting for 60.41%, among which the expression patterns of *metC* (*N134_RS01520*), *tycB* (*N134_RS01525*), *tycC* (*N134_RS01530*), and *opuC* (*N134_RS00920*) in the 3 groups were all in accordance with those of in RNA-seq. These results were deemed to be relatively reliable.

## 4. Conclusions

In conclusion, soybean peptides could promote DEGs from the acc and fab gene families as a critical role in fatty acid biosynthesis, while soybean protein modulated them oppositely, which was consistent with the results of organic acids secretion reported before. Both soybean protein and peptides improved gene expression enriched in purine biosynthesis and IMP biosynthesis pathways to provide more abundant nucleic acid and ATP for bacteria growth. Additionally, dpep instead of dpro induced genes enriched in ABC transporters pathway, promoted and prevented the accumulation of some amino acids in cells, and maintained the balance of osmotic gradient to improve the nutrition absorption. These results contribute to profound understanding of the mechanism of DEGs of *L. reuteri* LR08 regulated by soybean protein and peptides, as well as the relationship between physiological functions and gene expressions affected by dpep and dpro. The results of this study provide a new direction for the selection of nitrogen-sourced prebiotics in the food industry, as well as a new idea for the development of soybean protein products. Further research is needed to clarify how soybean protein and peptides affect specific gene expression within various pathways.

## Figures and Tables

**Figure 1 foods-11-01251-f001:**
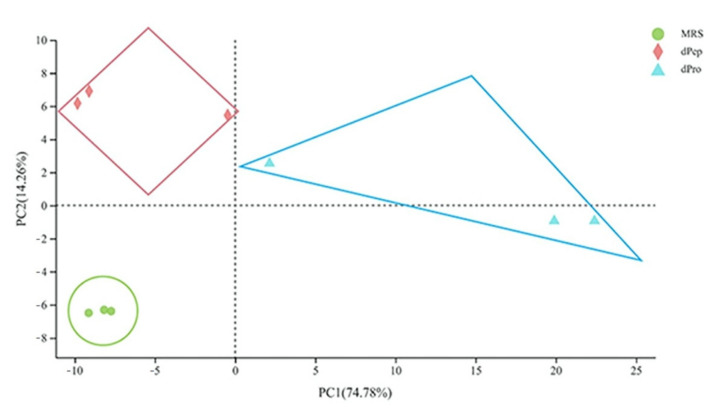
Principal Component Analysis of MRS, dpro and dpep groups (*n* = 3).

**Figure 2 foods-11-01251-f002:**
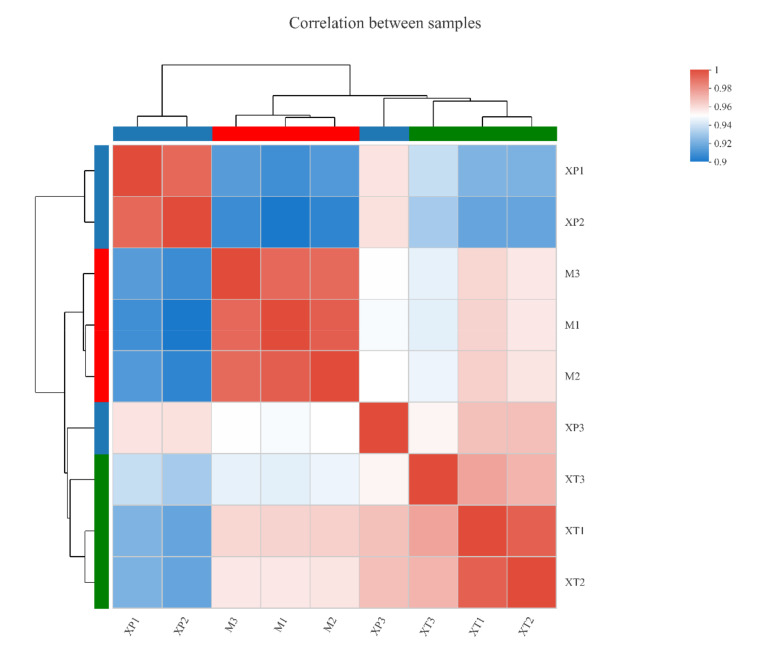
Correlation analysis between samples (*n* = 3). The names of samples are located on the right and lower sides of the figure, sample clustering situations are located on the left and upper sides, and squares of different colors represent the correlation between the two samples. M: MRS group; XT; dpep group; XP: dpro group.

**Figure 3 foods-11-01251-f003:**
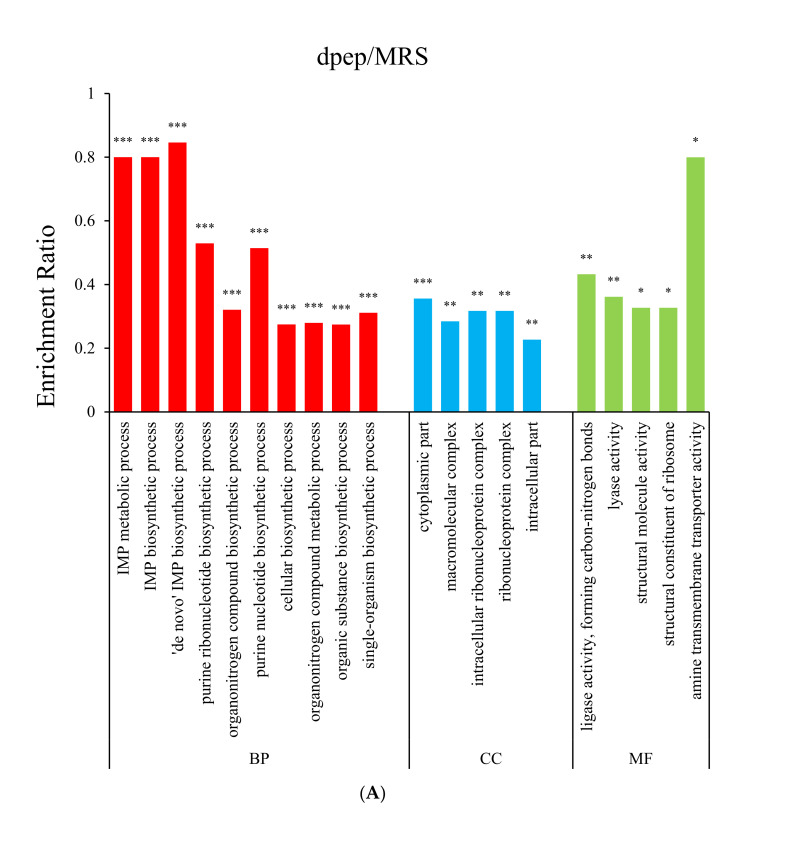
GO enrichment analysis of DEGs under different comparisons (*n* = 3): (**A**) dpep/MRS; (**B**) dpro/MRS; (**C**) dpep/dpro. BP: Biological process; CC: Cellular Component; MF: Molecular Function. Enrichment ratio refers to the number of genes annotated to the GO term in the gene set to the number of genes annotated to the GO term. The significance of enrichment is determined by the *p* value, in which *p* < 0.001, <0.01 and <0.05 marked as ***, **, and *, respectively.

**Figure 4 foods-11-01251-f004:**
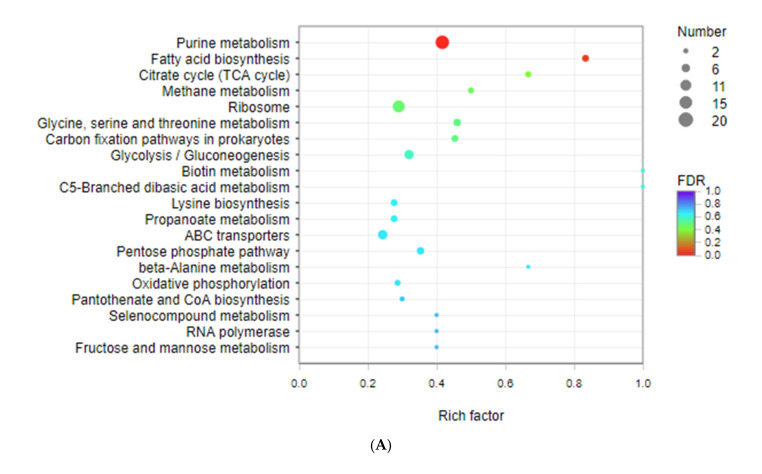
KEGG enrichment analysis of different comparisons (*n* = 3). (**A**) dpro/MRS; (**B**) dpep/MRS; (**C**) dpep/dpro. The vertical axis represents the name of the pathway and the horizontal axis represents the ratio of the Rich factor. The size of the dot indicates the number of genes enriched in this pathway, and the color corresponds to different Q value ranges.

**Figure 5 foods-11-01251-f005:**
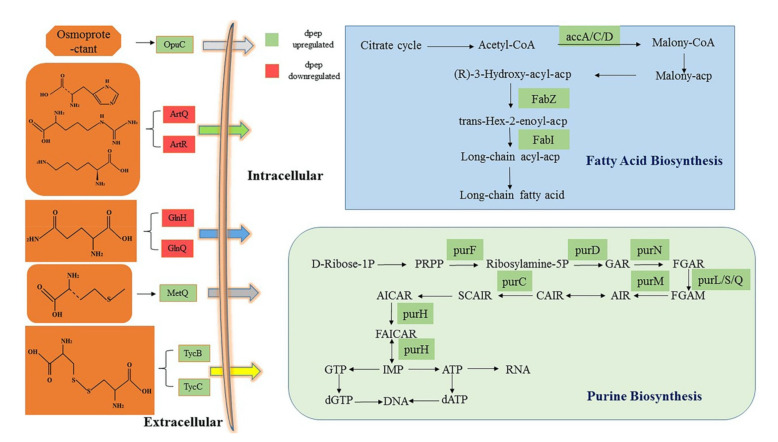
Schematic diagram of DEGs involved in crucial metabolism pathway of fatty acid, ABC transporters, purine biosynthesis in dpep/MRS, and depe/dpro comparison. Genes with green background represent up-regulation and red background represent down-regulation for dpep.

**Table 1 foods-11-01251-t001:** Composition and content of medium in different groups.

Composition	MRS (g/L)	dpep (L)	dpro (g/L)
**Peptone**	10	5	5
**Beef powder**	5	2.5	2.5
**Yeast extract powder**	4	2	2
**Dipotassium hydrogen phosphate**	2	2	2
**Anhydrous sodium acetate**	5	5	5
**Magnesium sulfate**	0.2	0.2	0.2
**Triamine citrate**	2	2	2
**Manganese sulfate**	0.05	0.05	0.05
**Twain 80**	1	1	1
**Glucose**	20	20	20
**Digested soybean peptides**	-	13.214	-
**Digested soybean protein**	-	-	12.259

## Data Availability

The data used or analyzed during the current study are available from the corresponding author on reasonable request. The data are not publicly available due to privacy.

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
