# Peer review of "Investigating Differential Expressed Genes of Limosilactobacillus reuteri LR08 Regulated by Soybean Protein and Peptides"

_foods, 2022, doi:10.3390/foods11091251_

Round 1

Reviewer 1 Report

The authors have absolutely nothing to describe the impact of their results on the food industry. What are the prospects for using the results in food technology. Basic information is missing.

The discussion of the results is presented in a chaotic manner. There is a lack of organization and sharpening of the results obtained. I think that non-reading authors move well with their thoughts between paragraphs, therefore the interpretation of the results is very difficult for the reader. There are no conclusions after presenting the results in the text. Why is this happening? How do the authors think etc.?

Gene names should be italicized.

There is no space between individual subsections.

The authors should check the statistical analysis as there are errors in figure 3

Lack of discussion of the results with the available scientific literature in some subsections disqualifies this article. It will take too long to rewrite the article. Prepare it properly responding to every suggestion and send it again.

Reviewer 2 Report

Soya is a traditional ingredient in feed and in food, the research focus is now searching alternative protein sources for feeding animal and humans. I am wondering why the authors tested soyben instead of other sources and /or why they did not include a control protein (other sources)?

the rationale of looking for novel  prebiotics it is interesting, but I expected a more "sustainable" approach. At least I suggest to cite relevant manuscripts which perform similar analysis with different protein source. I also suggest to discuss this point.

Table 1 and 2 . I suggest to merge and move to supplementary material

line 68. what do you mean? how did the author s perform the replacement?

Reviewer 3 Report

In this manuscript entitled "Investigating differential expressed genes of Limosilactobacillus reuteri LR08 regulated by soybean protein and peptides", the authors evaluate the effect of soybean protein and peptides on the differential expressed genes of Limosilactobacillus reuteri LR08 by the transcriptome analysis. Despite the intriguing observations, it contains several points that need to be corrected. I have comments, explained below. I hope that my comments are very useful for the improvement of this research.

Major comments
(1)    Protein and peptides: Information on soybean proteins and peptides added to Limosilactobacillus reuteri (L. reuteri) should be included. For example, nitrogen content, the molecular weight, and amino acid composition of soybean proteins and peptides are necessary.
(2)    dpro and dpep: Although done in the abstract, dpro and dpep are not defined in the main text.
(3)    MRS medium: It would be better to give a table showing the composition of the MRS medium when dpro and dpep are added. The way it is currently written, it is not clear which nitrogen source of the MRS medium was replaced with which dpro and dpep.
(4)    Bacteria: Since there is no information at all on the status of the L. reuteri that were subjected to transcriptome analysis, I think it is necessary to show the growth curves of the L. reuteri and indicate which phase (log phase, stationary phase, and so on) they were in.
(5)    Statistical Analysis: Please review the statistical method because it is incorrect; Authors used the Duncan’s multiple range tests as statistical analysis. But, Duncan's multiple range test has been pointed out to have problems such as not taking multiplicity. Thus, please change to another multiple tests. Duncan’s multiple range tests is very prone to significant differences, so using the correct method should result in no significant differences in many items.

Minor comments
(6)    L80: “Reverse transcription of cDNA” is uncomfortable. cDNA is made by reverse transcription of RNA. Therefore, I believe that expressions such as “Reverse transcription of RNA” are correct.
(7)    L92-93: This sentence should be indicated in the previous section as it relates to the RT.
(8)    L120-121: I don't know what you have determined that reliable data has been obtained. For example, I would need to indicate the number of total leads obtained by NGS, valid leads, etc.
(9)    Fig.2 : It would be good to transfer this data to a supplement.
(10)    L292-294: This information should be transferred to the section of “Materials and method”. In addition, I think this sentence means that you are measuring 17 internal standard genes, is that correct?
(11)    Table and Figure: The n number of data is not shown, please show it in the respective Table and Figure.

Round 2

Reviewer 1 Report

The abstract has not been altered at all as suggested in an earlier review. Add more information about the results received.

Graphical Abstract is so small that I can't judge anything. Do not duplicate the obtained results (graphs) in the graphic abstract. It makes no sense. Authors should show creativity, not copying. 

Citation: Lastname, F .; Lastname, F .; Lastname, F. Title. Foods 2022, 11, x. Https://doi.org/10.3390/xxxxx - this issue is not completed

Introduction is a very simple written language and does not introduce the reader to the presented topic of work. Authors should extend the Introduction description with the latest information again.

The authors have written nothing about food proteins, which have many nutritional and biological properties. After all, biologically active peptides are fragments of the amino acid sequences of food proteins that become active when released. Usually they are released during digestion, fermentation (thanks to the proteolytic activity of microorganisms and here the authors should extend the description of lactic acid bacteria) or in vitro enzymatic processes and then they can influence for human health. And such information should result from the presented scientific work. Add new references to increase the description of the lactic acid bacteria and their properties:
Chlebowska-Śmigiel, A., Kycia, K., Neffe-Skocińska, K., Kieliszek, M., Gniewosz, M., & Kołożyn-Krajewska, D. (2019). Effect of pullulan on physicochemical, microbiological, and sensory quality of yogurts. Current Pharmaceutical Biotechnology, 20 (6), 489-496. https://doi.org/10.2174/1389201020666190416151129
Chlebowska-Smigiel, A., Gniewosz, M., Kieliszek, M., & Bzducha-Wrobel, A. (2017). The effect of pullulan on the growth and acidifying activity of selected stool microflora of human. Current Pharmaceutical Biotechnology, 18 (2), 121-126. https://doi.org/10.2174/10.2174/1389201017666161229154324

It is worth the authors mentioning antimicrobial peptides. This is a very important and new issue that is of great importance in food technology. Despite intensive research, the mechanism of their action is not fully understood. The most important effects of these peptides are: changes in the permeability of the cell membrane, destabilization of its lipid structure, formation of micelles or channels in the membrane, binding with lipopolysaccharide, inhibition of DNA replication, inhibition of protein expression and ATP release, and in the next stage - cell lysis. The antimicrobial activity of the peptides towards microorganisms is also attributed to their ability to adopt the amphipathic α-helical structure. This information can be of great importance in the fermentation processes carried out by bacteria. The authors should mention this.

Figure 1 adds nothing to work. This graph should be deleted. The results from this issue are also not thoroughly discussed.

Figure 2 is not discussed in the text

The work lacks a description of the procedures for obtaining biologically active peptides with a potential preventive effect, they include the use of analytical and computer methods as well as proteomic and peptidomic techniques used in vitro, in vivo and in silico, including the processes of hydrolysis, separation, identification and determination of the activity of the obtained protein fragments along with the analysis of the relationship between the structure and biological function. The authors should describe this aspect.

All the statistical analysis is improving.

Reviewer 3 Report

I am satisfied with the revisions that have been made by the authors.

Author Response

Thanks for your previous questions and advice.